
**Ion acceleration at dipolarization fronts associated with**
**interchange instability in the magnetotail**
**Chao Sun[1], Yasong Ge[2], Haoyu Lu[1, 3]**
*[1]School of Space and Environment, Beihang University, Beijing,*
*100191, China (lvhy@buaa.edu.cn)*
*[2]Institute of Geology and Geophysics, Chinese Academy of Sciences,*
*Beijing, 100029, China*
*[3]Lunar and Planetary Science Laboratory, Macau University of*
*Science and Technology − Partner Laboratory of Key Laboratory of*
*Lunar and Deep Space Exploration, Chinese Academy of Sciences,*
*Macau, 519020, China*
**Abstrac**t It has been confirmed that dipolarization fronts (DFs) can be a
result from the existence of interchange instability in the magnetotail. In
this paper, we used a Hall MHD model to simulate the evolution of the
interchange instability, which produces DFs on the leading edge. A test
particle simulation was performed to study the physical phenomenon of
ion acceleration on DF. Numerical simulation indicates that almost all
particles move towards the earthward and dawnward and then drift to the
tail. The DF-reflected ion population on the duskside appears earlier as a
consequence of the asymmetric Hall electric field. Ions, with dawn-dusk



asymmetric semicircle behind the DF, may tend to be accelerated to a
higher energy (>13.5keV). These high-energy particles are eventually
concentrated in the dawnside. Ions experience effective acceleration by the
dawnward electric field Ey while they drift through the dawn flank of the
front towards the tail.



**Introduction**

Earthward moving high-speed plasma flows, which are called bursty bulk
flows (BBFs), play a vital important role in carrying significant amounts
of mass, energy, and magnetic flux from the reconnection region to the
near-Earth magnetotail (Angelopoulos et al., 1994). BBFs are often
accompanied with a strong (∼10nT), abrupt (several seconds), transient
enhancement of the magnetic field component Bz in the leading part,
known as a dipolarization front (DF) (Nakamura et al., 2009; Sergeev et
al.,2009; Fu et al., 2012a). Ahead of the DF, a minor Bz dip usually be
observed by THEMIS and MMS (Runov et al., 2009; Schmid et al., 2016),
which may be typically interpreted as strong diamagnetic currents caused
by a plasma pressure drop over the front or magnetic flux passing over
the SC or transient reconnection (Kiehas et al., 2009; Ge et al., 2011;
Schmid et al., 2011). Simulations have suggested that the magnetic
energy would be transferred to plasma on the DF layer in the Bz dip
region ahead of trailing fronts (Lu et al., 2017). Many of studies show
that the passage of a magnetic island (Ohtani et al., 2004), jet braking
(Birn et al., 2011), transient reconnection (Sitnov et al., 2009; Fu et al.,
2013), and/or the interchange/ballooning instability (Guzdar et al., 2010;
Pritchett and Coroniti, 2013) may account for DF generation. Both
Cluster and MMS observed that DFs propagate not only earthward but
also tailward, since the fast-moving DFs are compressed and reflected,



three quarters of the DFs propagate earthward and about one quarter
tailward (Zhou et al., 2011; Nakamura et al., 2013; Huang et al., 2015;
Schmid et al., 2016).
Spacecraft observations showed that the sudden energy increase in
charged particle fluxes at DFs from tens to hundreds of keV in the
magnetotail (Runov et al., 2011; Zhou et al., 2010; Fu et al., 2011; Li et
al., 2011; Artemyev et al., 2012). A series of studies have been conducted
to understand the signatures of DFs and particles, as well as the particle
acceleration mechanisms on the DFs. Li et al., (2011) studied the force
balance between the Maxwell tension and the total pressure gradient
surrounding the DF and found that the imbalance between the curvature
force density and the pressure gradient force density would lead to the
flux tube acceleration. Ions, essentially nonadiabatic in the magnetotail,
can be directly accelerated along the electric field produced by earthward
convection of the front, such as due to surfing acceleration or shock drift
acceleration (Birn et al., 2012, 2013; Ukhorskiy et al., 2013; Artemyev et
al., 2014). Electrons, comparatively adiabatic over most of their orbits,
can be accelerated through betatron and Fermi process (Birn et al., 2004,
2012). It is noticed that the magnetic field amplitude behind DF is much
greater than that ahead of it, Zhou et al. (2011, 2014) obtained that the
earthward moving front can reflect and accelerate ions. Ukhorskiy et al.
(2013, 2017) took the magnetic field component Bz for different areas



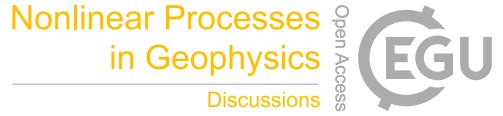

and situations into account, revealing a new robust acceleration
mechanism enabled by stable trapping of ions. In most cases, ions are
energized by combined actions from different acceleration mechanisms.
Nevertheless, the physical processes that generate suprathermal particles
are not yet fully understood.
On the simulation ground, previous two-dimensional simulations just
unveil large scale physical process concerning DFs, in their models, the
electric field (most of them are derived from $-V \times B$) is assumed to be
solely in the y direction behind DFs (Ukhorskiy et al., 2012; Greco et al.,
2014; Zhou et al., 2014). It has been found that the spatial scale of DFs in
the dawn-dusk direction is about 1-3 $R_E$ and its thickness is on the order
of the ion inertial length (Runov et al., 2011; Schmid et al., 2011), which
would be between 500 and 1000 km. In the sub-proton scale, there is an
electric field directed normal to the DF. The frozen-in condition is broken
at the DF and the electric field is mainly attributed by the Hall and
electron pressure gradient terms, with the Hall term dominants (Fu et al.,
2012b; Lu et al., 2013; Lu et al., 2015). Therefore, the Hall MHD model
is necessary to obtain the Hall electric field, which may determine the
electric system on DFs.
Lu et al. (2013) have successfully simulated the DF associated with
interchange instability in the magnetotail and the trend of simulated
physical variables are in good agreement with observations. In this paper,





we improves the simulation model in order to study how the Hall electric
field on DFs acts on the particle trajectories and ion energizations. Since
the DF is produced by temporal evolution of interchange instability
self-consistently, it would be meaningful to understand the ion
acceleration mechanism associated with the interchange instability in the
magnetotail.

**Theoretical and Numerical Model**
Numerical simulations have proved that the existence of interchange
instability triggered by the tailward gradient of thermal pressure and the
earthward magnetic curvature force is a possible generation mechanism
of the DFs in the magnetotail (Guzdar et al., 2010). Based on the Hall
MHD model associated with interchange instability (Lu et al., 2013), we
conducted test particle simulations to track ions trajectories backward in
time.
Our simulation was performed by two steps, the first is to establish a
more realistic DF to get particle motion background. The other is to place
test particles and track their trajectories.
The simulation coordinate system is defined with the x-axis pointing
away from the Earth, the y-axis pointing from dusk to dawn, and the
z-axis pointing northward (Guzdar et al., 2010, Figure 1). The breaking of
the earthward flow together with the curvature of the vertical field leads



to an effective gravity g away from the earth. Dimensional units are based
on a magnetic field of 15 nT, the Alfven velocity of 750 km/s, and
reference length of 1 $R_E$ leading to a time unit of ~8.5 s, an electric field
of 11.25 mV/m, and a pressure unit of 0.179 nPa.
The dimensionless model with an effective gravity is as follows:

$$
\frac{\partial}{\partial t}\begin{bmatrix} \rho \\ \rho\mathbf{U} \\ \mathbf{B} \\ \rho e_t \end{bmatrix} + \nabla\cdot\begin{bmatrix} \rho\mathbf{U} \\ \rho\mathbf{UU}+P\mathbf{I}\text{-}\dfrac{\mathbf{BB}}{\mu_m} \\ \mathbf{UB}\text{-}\mathbf{BU} \\ (\rho e_t+P)\mathbf{U}\text{-}\dfrac{\mathbf{B}}{\mu_m}(\mathbf{U}\cdot\mathbf{B}) \end{bmatrix} = \begin{bmatrix} 0 \\ g \\ 0 \\ g.U \end{bmatrix} +
$$

$$
d_i\begin{bmatrix} 0 \\ 0 \\ -\dfrac{1}{\mu_0}\nabla\times\left(\dfrac{\nabla\times\mathbf{B}\times\mathbf{B}}{\rho}\right) \\ -\dfrac{1}{\mu_0^2}\mathbf{B}\cdot\left[\nabla\times\left(\dfrac{\nabla\times\mathbf{B}\times\mathbf{B}}{\rho}\right)\right] \end{bmatrix} + d_i\begin{bmatrix} 0 \\ 0 \\ -\dfrac{1}{\mu_0}\nabla\times\left(\dfrac{\nabla\rho_e}{\rho}\right) \\ -\dfrac{1}{\mu_0^2}\mathbf{B}\cdot\left[\nabla\times\left(\dfrac{\nabla\rho_e}{\rho}\right)\right] \end{bmatrix}
$$

(1)

Where $P = p + \boldsymbol{B}^2/2\mu_0$, $\mathbf{U}$ and $\mathbf{B}$ are velocity vector and magnetic
field vector, respectively, $\rho e_t = \rho\boldsymbol{U}^2/2 + p/(\gamma-1) + \boldsymbol{B}^2/2\mu_0$, β is
plasma beta, $g_x$ is the effective gravitational force in x direction. In
equation (1), the second and third terms on the right-hand side represent
the Hall effect and the electron pressure gradient, respectively. In our
present numerical cases, we postulate that plasma is under isothermal
conditions with an isothermal equation of state $p = \beta\rho/2$ and take the
adiabatic    exponent    $\gamma = 5/3$ .    The    ion    inertial    length    $d_i =$
$(m_i/\mu_0 e^2 Z^2 L^2 n_i)^{1/2}$ , given the reference length L = 1 $R_E$, the
dimensionless ion inertial length is taken as $d_i \approx 0.1$. Electron pressure



$p_e$ is taken as $p/6$, because the proton temperature is 5 times that of
electron temperature (Baumjohann et al., 1989; Artemyev et al., 2011).
As for initial conditions, the quasi-stationary equilibrium built by the
plasma pressure and effective gravity g (see equation (2)) (Guzdar et al.
2010 and Lu et al. 2013, 2015) is theoretically reasonable.
$$\hat{g}\frac{\beta}{2} = \frac{\partial}{\partial x}\left(\frac{\beta}{2}\rho + \frac{B_z^2}{2}\right) \tag{2}$$

It should be noticed that the dawn-dusk and earthward electric field
components averagely, increase to ~5 mV/m along with the transient Bz
increase and in some events, the electric field increase exceeded 10
mV/m (Runov et al., 2009, 2011; Schmid, D., et al. 2016). However, the
electric fields calculated by the Hall MHD model in Lu et al. (2013) are
smaller than the observations (see Lu et al., 2013, for a typical
dipolarization event at x = -10 $R_E$ in the equatorial plane, we set $B_0$ = 15
nT, leading to Bz changed from 10.2 nT to 16.8 nT after DF propagation.
The electric field components Ex and Ey are both less than 3 mV/m). So,
it is reasonable that we improve the initial conditions to obtain a realistic
electric field, which plays a vital important role in ion energization.
We take the initial conditions as follows:
$$\rho(\mathrm{x}) = \frac{1}{2}(\rho_L + \rho_R) - \frac{1}{2}(\rho_L - \rho_R)\tanh\left(\frac{x}{l}\right) \tag{3}$$

$$\begin{cases} B_Z(x) = 0.28x + 0.7535 & (x \le -0.38) \\ B_Z(x) = 1.5 + \tanh\left(\frac{x}{0.3}\right) & (-0.38 < x < 0.4) \\ B_Z(x) = 0.14x + 2.314 & (x \ge 0.4) \end{cases} \tag{4}$$

Given the generalized Ohm's law, we use a piecewise function to

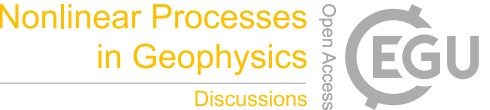

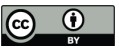

describe Bz so as to obtain a strong electric field. In equation (3), $\rho_L$
and $\rho_R$ are the density closer to and away from the Earth, respectively
and the characteristic scale $l = 0.2\ R_E$.
We solved equation (1) by adopting the second-order upwind total
variation diminishing scheme. The simulation box is $2\ R_E$ and $1.5\ R_E$ in
the direction of x and y, respectively. The x boundary is assumed to be
zero for all perturbed quantities and the y boundary is to be periodic.
As the second simulation step, the control equations for ion motion
should be given. Typically, the drift approximation breaks down in terms
of ion motion in magnetotail. The dimensionless equations of motion are
given by
$$\begin{cases} \dfrac{d\boldsymbol{r}}{dt} = \boldsymbol{u} \\ \dfrac{d\boldsymbol{u}}{dt} = \alpha(\boldsymbol{E} + \boldsymbol{V} \times \boldsymbol{B}) \end{cases} \tag{5}$$

where $\boldsymbol{r}$ is the particle position, $\boldsymbol{u}$ is the particle velocity, the
dimensional parameter $\alpha = \dfrac{1}{d_i} \approx 10$.

**Simulation Results**
From 0s to 144.5s, the simulation experienced a pre-onset phase, during
which the DF formed as a consequence of effective gravity g interaction
with plasma density gradient. In order to be more realistic, we set up the
time interval from 144.5s to 187s as the acceleration period of the
particles.

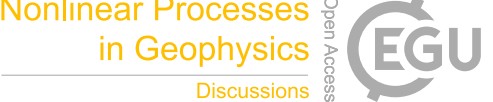

Figure 1 shows the evolution of the electric field in the z = 0 plane and
black lines indicate streamlines, one can clearly see that the DF moves
toward the Earth as time passes by. From Figure 1b it can be seen that the
earthward flows coexisted with the tailward flows of the dawn and dusk
edges, as a consequence two vortex flow pattern appeared. Figure 1 also
shows that the electric field components Ex and Ey are both normal to the
front, which is consistent with the observation and simulation (Fu et al.,
2012b; Lu et al., 2013). It should be noticed that the electric field is
asymmetrically distributed on the DF, with a stronger dawnside electric
field. This asymmetry can be interpreted that the two vortexes produce
the convection electric field in the direction of dusk-dawn, which
generate superposition and cancellation of the dawn and dusk side electric
field of DF respectively.



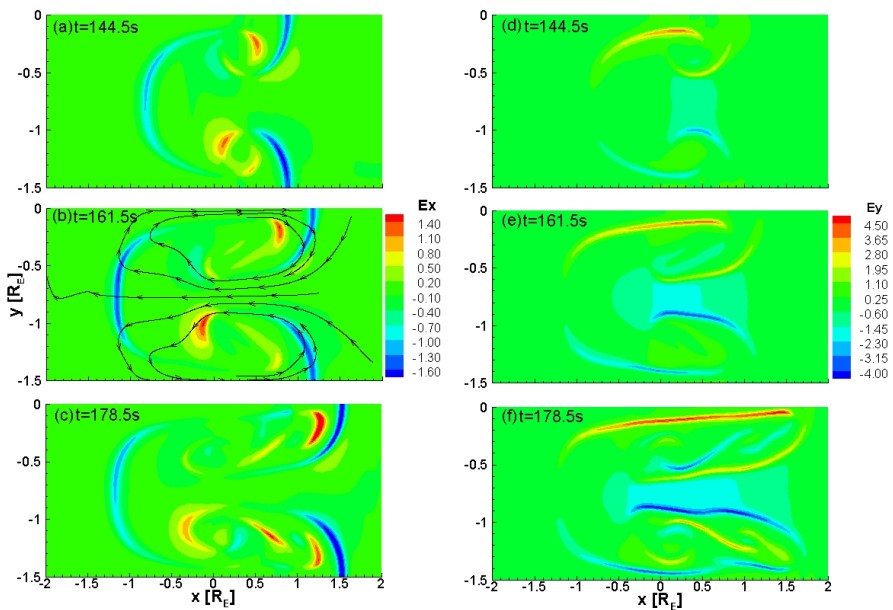

**Figure 1.** Evolution of the electric field Ex (a-c) and Ey (d-f), black line

in (b) indicate streamlines

At t = 144.5s, we numerically distribute test particles (80000 ions in total)

around the DF (a simulation box with x = -0.9 $R_E$ ~ -0.4 $R_E$, y = -1.46 $R_E$

~ -0.04 $R_E$) with the initial power law energy distribution F~$(1 +$

$h/\kappa T_0)^{-\kappa-1}$(we take  κ = 5, $T_0 = 1.5\ keV$  and h from 1 keV to 10 keV)

( Artemyev et al., 2015). Figure 2 exhibits the spatial distribution of

protons at a given moment. The energy of particles is marked with color

and black lines indicate the position of DFs. As time passes by, the ions

behind the DF accelerate and transport to the dawn flank of the DF,

resulting in the reduction of the ion density behind the DF. We investigated

the characteristics of ions trajectories and found that the behaviors of ions

consist of two parts (not shown), one is forced by the electric field Ex at

the leading edge of the front, resulting in earthward motion and
dawnward drift. Another is due to the electric field Ey at the dawn flank
of the DF, leading to tailward drift. These results are consistent with
observations and simulations (Nakamura et al., 2002; Greco et al., 2014;
Zhou et al., 2011, 2014). Therefore, the electric field on the DF (Figure
1a), mainly produced by Hall term and always normal to the front (Fu et
al., 2012b; Lu et al., 2013), makes the particles move in the way
described above. Statistical analysis of the ions energy in Figure 2
indicates that the maximum energy is about 27 keV. In order to better
distinguish particles from different energy, we assumed that the ions with
the final energy greater than 13.5 keV are high-energy particles.

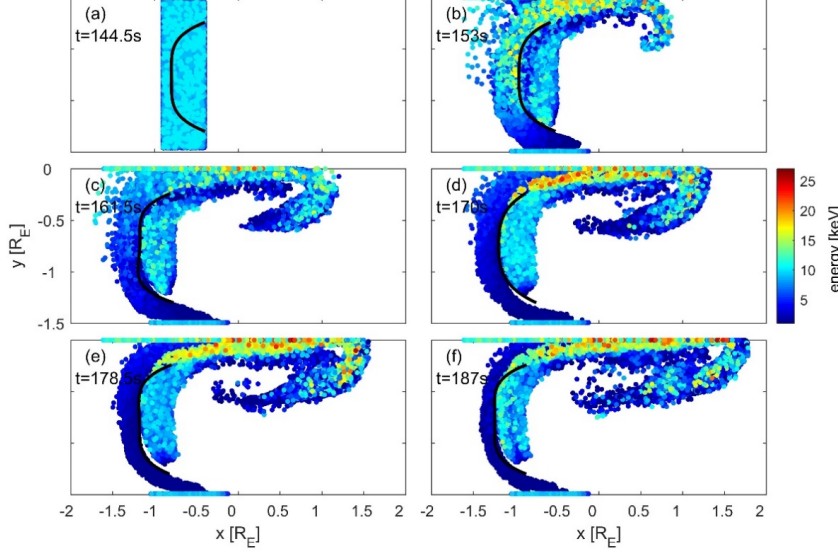


**Figure 2.** Test particle simulations of proton energization at the DF,
particle energy is indicated with color and black line represents the
position of the DF

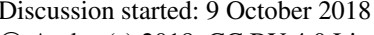

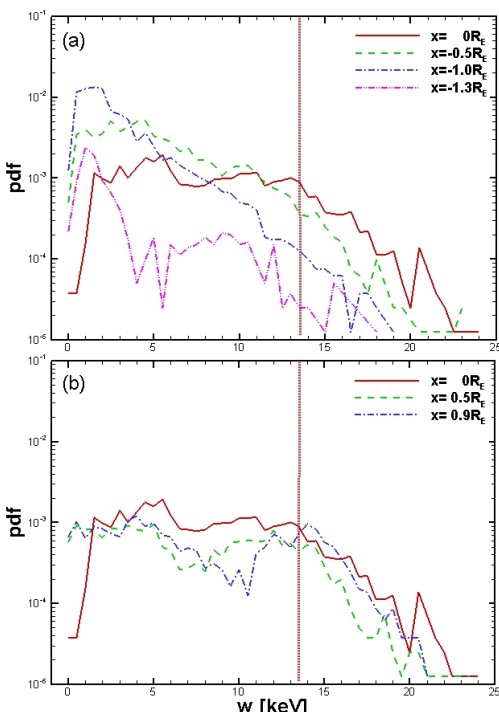


**Figure 3.** PDFs of particle energy computed at the region of (a) x < 0 $R_E$

and (b) x > 0 $R_E$. The red dotted line mark the high energy demarcation

line 13.5 keV.

Figure 3 gives the probability density function (PDF) of particle energy at

different x positions. In order to better distinguish the curves of different

x distances among the multiple fold lines, we show the results in two

figures according to different region in x direction. It can be seen from

Figure 3b that the high-energy particles are assembled in the region of x >

-0.5 $R_E$ whereas Figure 3a shows that the small energy (~ 2keV) ions are

concentrated in the region of x < -0.5 $R_E$. At x = 0 $R_E$, ion energy is

evenly distributed between 2 keV and 16 keV practically. In combination

with Figure 2, we can further obtain that almost all the high-energy
particles gathered in the dawnside of x > -0.5 $R_E$ region. It implies that
ion acceleration is more effective at the dawnside of DF.
To have a statistical description of high-energy ions, we picked out
high-energy particles from the total number. The simulation results are
shown in Figure 4 with ions energy marked with different color. It
appears that high energy particles, accounting for 6 percent, mainly
gathering at the dawnside of the DF.

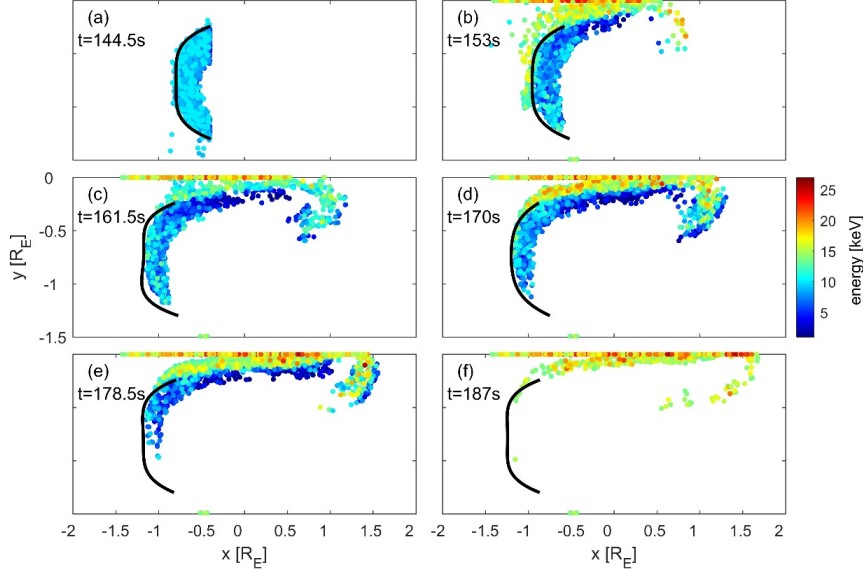


**Figure 4.** Snapshots of high-energy ions at specific moment of the
simulation, black line represents the position of the DF
Figure 4a shows that the initial position of high-energy particles is
roughly an asymmetric semicircle whereas the dawnside area is wider
than the duskside, which means that more ions are accelerated in the

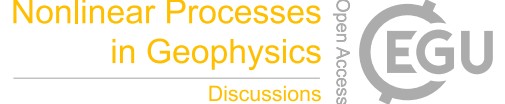



dawnside than in the duskside. Compared with Figure 2, however, we can
intuitively infer that ions with initial positions ahead of or behind and
away from the front would not obtain great energization. The ions with
initial positions ahead of the front are forced by the Ex of pre-DF region
and they move earthward and dawnward with a larger gyration radius due
to smaller ambient magnetic Bz, thus can't be accelerated by the
dawnside electric field Ey. The ions with initial positions behind the front
move with it and always stay behind the DF during the whole evolution
period of DF. As a result, there exist no strong fields to energized ions.
That is to say, only particles which diverted to the dawnside region closer
to the front can be effectively accelerated.
In a previous paper, Zhou et al. (2014) inferred that the more energized
DF-reflected ions originating from the duskside of the DF would be able
to reach farther into the ambient. In their model the ions would have been
accelerated more significantly in the DF duskside than in its dawnside
which is due to the y displacements behind the convex DF (Zhou et al.,
2014 Figure 3). However, observations and numerical simulations
indicate that the convective electric field behind the front is smaller than
the Hall term on the DF on the spatial scale of ion inertial length(Fu et al.,
2012b). Therefore, the explanation based on the convective electric field
Ey was inappropriate in oue model. Figure 4 has already illustrated that
the ion acceleration process is on the dawnside. In addition, statistical



analysis of 4863 high-energy ions indicates that 1570 ions were traced to
the duskside of the DF, about 32 % of the total high-energy particles. The
source area of ions reaches closer to the Earth, as shown in Figure5.

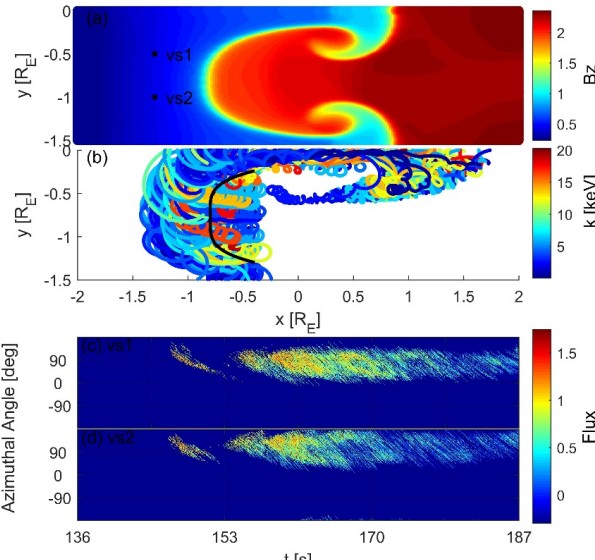


**Figure 5.** Simulation results of ion differential energy fluxes in the 1-20
keV energy range at different location. (a) Positions of virtual satellites.
(b) Ions with initial positions on different y distances moving with the
dipolarization front, the DF at t = 144.5 s was marked with black solid
line. Kinetic energy at finial moment is marked with color. (c and d)
Energy fluxes at dawnside (vs1) and duskside (vs2), respectively, as the
functions of equatorial azimuthal angle and time.
The dark spots in Figure 5a mark the locations of the virtual satellites.
Figure 5c and 5d show the distribution of differential energy flux as the
function of equatorial azimuthal angle and time at the duskside and
dawnside of the DF respectively. Ion trajectories with initial positions





along different y distances in Figure 5c and 5d are plotted in Figure 5b
where the DF at t = 144.5 s is set as baseline and marked with black solid
line. Kinetic energy at finial moment is indicated with color.
It is obviously seen in Figure 5 that the duskside ions tend to move to
dawn at the front (Figure 5d, $90° < \theta < 180°$), while the dawnside ones
divert toward tailward (Figure 5c, $0° < \theta < 90°$). This finding is similar
to the fluxes of 78-300 keV protons in Birn et al. (2015). At about t =
146s ~153s, the particles with higher initial energy ahead of the front
have large radius of gyration and those particles are minor affected by the
smaller initial electric field, therefore they are almost simultaneously
observed (Figure 5b and 5d). While at t > 153s ions with the initial
position at duskside would be able to reach farther into the ambient,
which is consistent with the results of Zhou et al. (2014) and Birn et al.
(2015). On the other hand, the earlier observed ions are not the most
energized ions compared with high-energy counterparts in our model,
which is opposite to Zhou's conclusion. It can be easily understood by
considering the Hall electric field. The small electric field near the
duskside of the front allows particles to drift toward earthward and
dawnward for a long time, whereas the high one close to dawnside forces
ions to drift tailward quickly during the period that particles obtain most
energy (Figure 4).
In order to study how the Hall field Ey on the dawnside of DF accelerate


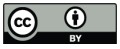

ions, we choose one typical ion to track its trajectory, which is initiated at
x = -0.7 $R_E$, y = -0.86 $R_E$ behind the DF with an azimuth angle of 14.58°
and initial kinetic energy of 1 keV. Its final kinetic energy is 12 keV, as
shown in Figure 6. Figure 7 demonstrates the evolution of ion positions
and energy. During the beginning period from 144.5s to 162s, the ion
moves earthward together with the front and meanwhile dawnward in the
frame of the moving front. During this period, the ion gains very little
energy. Even though the Ex component of electric field accelerates the
ion along its earthward motion, the deceleration by the Ey component
keeps the ion energy almost unchanged. When t = 163.2 s, the ion arrives
at the dawnside of the DF, where the Hall electric field is very strong.
After a sharp energization for about 3 seconds, the ion kinetic energy
increase to ~ 10 keV (Figure 7b and 7c, the weaker Ex works to reduce
the energy by about 8 keV and the stronger Ey increases the energy by
about 20 keV). As shown in Figure 6 and 7 that after t > 166 s the ion
kinetic energy gradually increases, which can be interpreted that the
y-displacement $\delta y^+$ (corresponding to the energy increase) is larger than
$\delta y^-$ (corresponding to the energy reduction) in the case where Ey
component is almost constant.



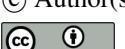


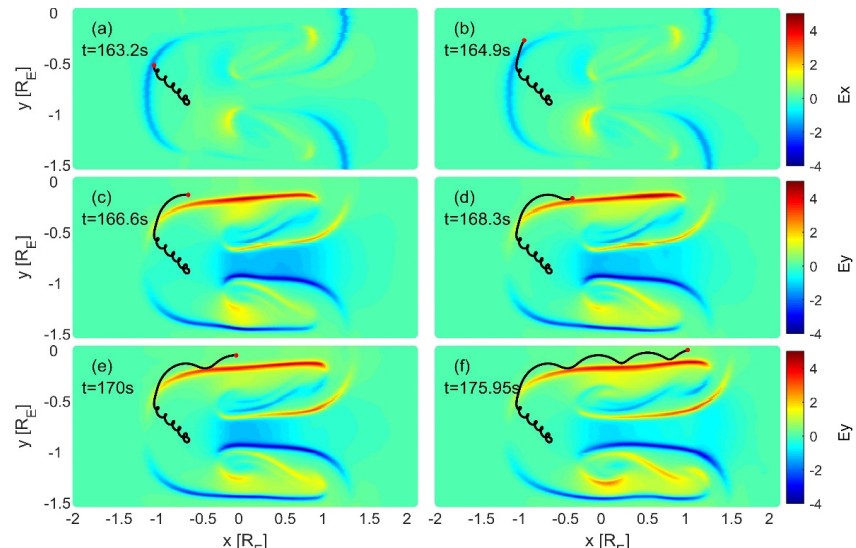

**Figure 6.** Orbits of a proton with the initial energy 1 keV and final energy

12 keV, traced from x = -0.7 $R_E$, y = -0.86 $R_E$ at different moments. The

locations of proton are shown as red dots superposed on snapshots of the

background Hall electric field Ex (a-b) and Ey (c-f).

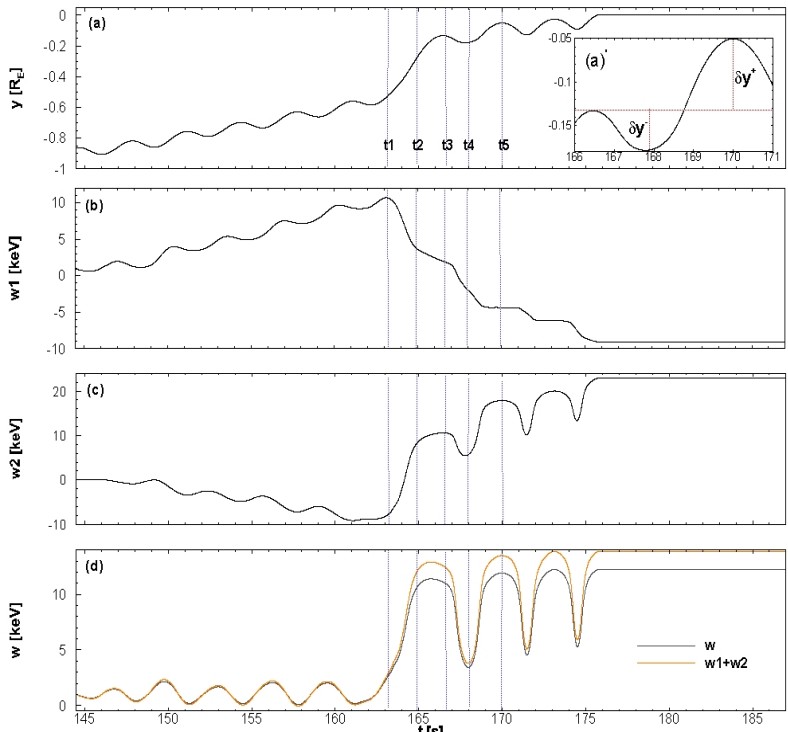

326

**Figure 7.** Physical quantities of ion as the function of time with blue
dotted lines index specific moment. (a), (a') Y position and its partial
enlarged detail, red dotted line is the reference line. (b, c) Energization
produced by Ex and Ey, respectively. (d) Kinetic energy and numerical
summation of w1 and w2 display with orange and black line, respectively.
The label of t1 to t5 correspond to 163.2s, 164.9s, 166.6s, 168s and 170s
respectively.

**Summary and Discussion**

In this paper, we used a test particle simulation to investigate ion
acceleration at dipolarization fronts (DFs) produced by interchange



instability in the magnetotail, by performing a Hall MHD simulation. The
Hall MHD model was improved by applying the realistic initial
conditions to obtain the fields which are better consistent with
observation.
Test particles were settled in both the pre-DF and post-DF region, most of
them exhibited earthward and dawnward drift and then diverted tailward.
It is found that ions with the initial position at duskside would be able to
reach farther into the ambient plasma, which has been also proofed by
Zhou et al. (2014) and Birn et al. (2015). Statistical analysis shows that
the high-energy particles are mainly assembled in the dawnside of x >
-0.5 $R_E$ region, which suggests the dawnside region of the DF is the main
area for particle acceleration.
Numerical simulation results indicate that the ions initially settled behind
the front may obtain higher energization. In order to explain how the Hall
electric field influence ions, we tracked the trajectory of particular ions in
the ion-scale electric field. As expected, the Ey component at the dawn
flank of DF plays an important role in the acceleration of ion. Although
the Ex component in the pre-DF region constitutes a potential drop of ~ 1
keV across the DF as reported by Fu et al., (2012b), the energy
enhancement would be offset on their way out toward the magnetotail due
to the Ey component. The spatial and temporal properties of Ey
component are critical factors for particle acceleration (Greco et al., 2014;




Birn et al., 2013, 2015; Artemyev et al., 2015; Ukhorskiy et al., 2017). In
contrast to the results from other MHD model, it makes sense in our
self-consistent Hall MHD simulation that the accelerating electric field is
the Ey component of the Hall electric field on the dawnside of the front
instead of the convection electric field Ey behind the front in their model.
Our two-dimensional Hall MHD model can well reproduce the direct
acceleration process generated by the Hall field. Nevertheless, it should
be pointed out that the ion acceleration mechanisms such as Fermi
acceleration and resonance acceleration can also provide powerful ion
energization with tens of keV to hundreds of keV (Fu et al.,2011;
Artemyev et al., 2012), which is not discussed in this paper. Still, there is
no doubt that our study suggests that the dawn flank dusk-dawn electric
field plays an essential role in ions energization.

**Acknowledgements**
Our work was supported by the National Natural Science Foundation of
China (NSFC) under grants 41474144, 41674176, and 41474124, and the
fund of the Lunar and Planetary Science Laboratory, Macau University of
Science and Technology − Partner Laboratory of Key Laboratory of
Lunar and Deep Space Exploration, Chinese Academy of Sciences
(FDCT No. 039/2013/A2). The simulation data will be made available up
on request by contacting Haoyu Lu.

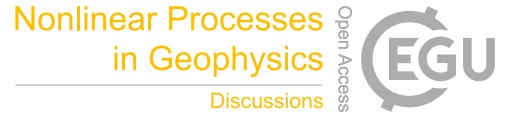

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
