# Peer review of "Discussion started: 9 October 2018 © Author(s) 2018. CC BY 4.0 License."

_Nonlinear Processes in Geophysics, 2018_

## Referee Comment (RC1) · Anonymous Referee #1 · 9 Nov 2018

[10pt]article

color,soul [margin=1.4in]geometry

Reviewing Comments on Manuscript: "Ion acceleration at dipolarization fronts associated with interchange instability in the magnetotail" by Sun et al.

**General comments:**

The authors studied ion acceleration at a diporlarization front (DF) produced by interchange instability in the magnetotail. They used a 2D Hall MHD model to simulate the time evolution of a DF associated with interchange instability and launched test particles in the electromagnetic fields from the Hall MHD model to investigate the ion acceleration. Initially, the test particles are launched around the moving DF. They found that the ions initially settled behind the DF obtain higher energization. The resulted high-energy ions ($> 13.5$ keV) are mainly assembled in the dawnside region of the DF. They conclude that the dawn-dusk component of the Hall electric field on the dawnside of the DF play a important role in the ion energization.

Dipolarization front has been found as an important transient structure contributing to the energy, plasma and magnetic flux transport in the near-Earth magnetotail during substorms based on observations and simulations. Plasma acceleration associated with the DFs is an important component to understand the magnetotail dynamics. Although the Hall electric field related to the DFs has been studied in previous observations and simulations, its effects on the ion acceleration are not well understood. This work investigates the relevent mechanism for the ion acceleration near the DFs. Therefore, it is important for the space physics community and appropriate for the journal. However, there are some comments needed to be addressed properly before publication.

**Major comments:**

(a) I have trouble to understand the initial conditions of the Hall MHD simulation (Lines 150-156). The authors claim that "we improve the initial conditions to obtain a realistic electric field" (Lines 148-149) in order to compare with the electric field (normal to the DF) which increases around 10 mV/m at the DFs in some observations. From Figure 1, we can see the $E_y$ has much larger values than $E_x$ at the flank and its peak value is 4.5 (normalized unit) which is 50.6 mV/m. Even at just the dawnside of the DF (e.g., x=-1.0 and y=-0.3 in Figure 1e), $E_y$ is about 2.0 (yellow color) in normalized unit, which is 22.5 mV/m. Thus, the magnitude of the electric field (i.e., the E field normal to the DF) at this location is even bigger than this value by including $E_x$. Such value is way much larger than observed 10 mV/m in many literature (Runov et al., 2009, 2011; Fu et al, 2012b). In this manuscript, the authors conclude that the ions are mainly accelerated on the dawnside of the DF and reach energy $> 13.5$ keV. In my opinion, such acceleration may be true but the energy gain by these ions may be overestimated. In addition, according to the Equation (4), there is a signifcant $B_z$ gradient in x direction in the entire simulation box. This could lead to a grad-B drift for the ions to the dawnside of the DF, which is related to the conclusions of this manuscript. Moreover, the transient half-width for the mass density is 0.2 in Equation (3), while that for the $B_z$ is 0.3 in Equation (4). This may lead to artificial local non-equilibrium which may not appear in the observations. Therefore, in general, I hope the authors can provide more discussions about how they select the initial conditions and why these conditions are reasonable.

(b) As far as I understand, the test particle simulation in this manuscript is based on the electromagnetic field from a two-dimensional (2D) Hall MHD simulation. Therefore, the test particle simulation is also 2D without the motion in z direction along the field line. In the real magnetotail condition (3D), for the case that the ions are convecting adiabatically with the magnetic field lines and bouncing along the z direction, this 2D simulation may be fine for representing the ion behaviors by projecting ion gyration motion onto this 2D plane. However, in the real magnetotail condition, once the ions experience non-adiabatic acceleration, they may not follow initial field lines and with the velocity in z, they may not bounce back to anywhere close to the initial field lines. This could be the situation that when the ions are getting acceleration by the Hall electric field. In this study, by using the 2D simulation, the authors investigate the ions acceleration by the Hall electric field on the dawnside of the DF and they show that the accelerated ions would assemble on the dawnside and get accelerated

multiple times (e.g., Figure 6 and 7). In reality, this may be true for the ions with 90-degree pitch angle, but for other ions, they may leave that region after the acceleration at the first time. Therefore, in my opinion, the authors should discuss this limitation and add such warning to their conclusions.

(c) As we know, the Hall electric field is usually formed by the relative bahaviors between ions and electrons at certain spatial scales. It is very sensitive to the electon and ion kinetics. Thus, theoretically, the ion kinetic behavior due to Hall electric field is better to study by using self-consistent kinetic models (e.g., Particle-in-Cell simulation). In this study, the Hall electric field is generated in a Hall MHD model. However, the corresponding ion kinetic behaviors in the regions with this Hall electric field are unknown. Launching test particles is equivalent to assume a state of ion kinetic behaviors (but it doesn't mean it happens in reality). To make this clear, let's take a look at the results in this manuscript. The authors show that most of the ions are drift to the dawnside of the DF and then drift tailward at the flank. This would lead to a dawnward and tailward current, which is supposed to reduce the Hall electric field self-consistently. Therefore, in my opinion, the authors should provide some discussions about such limitations.

**Minor comments:**

19-20: "all particles". This term is not well defined. The launch information (e.g., initial energy, locations, ...) of these test particles is necessary.

26: "Ey". Please use subscript "y". Same for all the "Bx", "By", "Ex" in the text of this manuscript.

40: "magnetic flux". Does it mean "magnetic flux tube" here?

41: "SC". This abbreviation has not been defined yet.

44-48: There are many possible mechanisms related to the DF generation. As a reader, I

would like to know, what is the difference of the electromagnetic system of the DF generated by interchange instability than other mechanisms?

108: "ions trajectories" → "ion trajectories"

108: "track ions trajectories backward in time". This phrase makes a little confusion. It could mean either running the simulation with negative time step from later time to earlier time or running the simulation with positive time step then check the time history of the trajectories of selected particles. Please specify which scenario is used in this manuscript.

111: "realistic". This word is not used properly. The simulation in this study is only two dimensional and the parameters are not set up based on spacecraft observations for specific events.

122: (1) "$\mu_m$" doesn't look correct in the second and fourth equations. (2) The "g" should be vector "**g**". (3)The $\nabla \rho_e$ term in the third and fourth equations. This should be derived from the electron pressure gradient $\nabla p_e$. Even though the authors consider isothermal conditions, there should be a temperature factor multiplying to the $\nabla \rho_e$ term.

124: "$\beta$". There is no an explicit "$\beta$" shown before this sentence.

125: "$g_x$". There is no an explicit "$g_x$" shown before this sentence. Instead, the authors should provide information about "**g**" and how to determine the value of **g** and to make it comparable to the observations.

129: What is the value of "$\beta$" used?

138: According to Guzdar et al. (2010), there is a missing "$\rho$" on the left hand side of the equation.

145: "we set" should be "they set".

154-155: What are the values for $\rho_L$ and $\rho_R$?

158-159: (1) "The simulation box is $2R_E$ and $1.5R_E$ in the direction of x and y". This description doesn't look correct. The simulation results (i.e., the figures) show that there is $4R_E$ in x. (2) Since the simulation used in this study is two-dimensional, please mention this information in the Abstract and somewhere in the "Theoretical and Numerical Model" section. (3) Please provide the information about the number of grid cells in each dimension.

161-167: As a reader, I would like to know more details about the test particle simulation: (1) Time step (2) The minimum gyration period of the simulated ions (3) Time integration scheme (4) What happens if the test particle hits the boundary of the simulation box? (5) Since the electromagnetic field is from a two-dimensional Hall MHD simulation, is the test particle simulation also two-dimensional (i.e., do not consider the motion along the z direction)? If so, the test particles should only represent the particles with 90-degree pitch angle.

172: "more realistic". It is not clear in what aspects this is more realistic. Please specify.

176: "DF". The DF is commonly defined by the abrupt increase of $B_z$ component. However, Figure 1 doesn't show any information about $B_z$. Please point out in Figure 1 where is DF and how it is defined.

182-184: "It should be ...... with a stronger dawnside electric field." It is not clear which asymmetry is mentioned in this sentence. Which electric field component? What time and location in Figure 1?

184-187: Please specify the locations with either Hall electric field or convective electric field

in Figure 1. Does the asymmetry include the Hall electric field? If so, it doesn't make sense to say that the convection electric field can cancel the Hall electric field. At a location, if the Hall effect is strong, then the electric field is explained as Hall electric field; if the Hall effect is weak, the electric field is understood as convection electric field.

192: "simulation box". Should it be "launch region of the test particles", since the results show that some of the particles are located outside this box? In addition, please also provide the DF location in text related to this launch region.

196: "protons". In this manuscript, both "proton" and "ion" are used. Since there is no any other ion species than proton, it should be better to only use one of them.

197: "black lines indicate the position of DFs". Are the black lines schematic or calculated? What variable is used to determine the black lines? Same comments for Figures 4 and 5b. In addition, what does the $B_z$ profile look like in the launch region? It would be helpful for comparing with observations if there is a line plot of $B_z$ as a function of x.

Figure 2: (1) (Same for Figure 4) What happens to the particles hitting the simulation boundary in y direction? Are they still moving (e.g., some energetic particles at y=0 in (b) is not there any more in (c)) (2) Why are there more energetic particles in earlier time (b) than those in later time (c)? Do they lose energy? (3) Why do most of the particles even with a distance in front of the DF (the black line) move to the dawnside? Why is there no accelerated ions in front of the DF due to the reflection by the moving DF in Panel (d)-(f) (e.g., Zhou et al., 2011)? (4) In Panels (c)-(f), there are large amount of particles with very low energy at the region with x=0-0.5 and y=-0.5. How do they reach that area earlier than all those energetic particles? (5) Behind the DF, from (b) to (d), there is an obivous increase amount of particles with energy of 10keV. Is there any local acceleration happening? And why do those particles become fewer in (e) and (f)?

Figure 3: (1) What time is this plot taken? (2) It should be better to add the initial PDFs so

that the initial power law can also be compared. (3) What are the areas to select the particles for plotting each line at different x location? Do these PDF curves include the particles at the y boundary? How much percentage of the energetic particles are at the boundary? If the number is significant, it should be better to perform a simulation with a bigger box.

223: "among the multiple fold lines". It is not quite clear what it means. Please rephrase it.

226: "small" → "low"

247-248: "they move earthward and dawnward with a larger gyration radius due to smaller ambient magnetic Bz." This is about the ions with initial positions ahead of the front. The dawnward motion is not explained well. Is the gyration radius comparable to the scale of the DF? Is the grad-B drift due to the set-up of the Hall MHD magnetic field considered?

277: "distribution of differential energy flux". How large area is used to select the particles for the calculation? In addition, please change the label of Figures 5c and d to "differential energy flux" instead of "flux".

278: "azimuthal angle". Which direction is indicated by the zero degree? Please define the directions indicated by different azimuthal angles.

279-280: "Ion trajectories with initial positions along different y distances". The meaning of this sentence is not quite clear (e.g., the y distance from where?), please rephrase it. In addition, it is very hard to obtain information from Figure 5b because too many lines overlapping to each other (e.g., can't find the initial position, can't follow individual trajectory,...).

286-290: "At about t=146 ...... they are almost simultaneously observed (Figure 5b and 5d)". The meaning of this sentence is not clear, please rephrase it. In addition, why is there a gap around t=153s in Figures 5c and d?

Figure 6: Figure 7 shows that the particle obtains large amount of energy from w2 between t=163-165s. It should be better to also show $E_y$ during this time period in Figure 6.

316-320: The explanation of the gradual increase of the kinetic energy is not clear. Because the magnitude of $\delta y^-$ and $\delta y^+$ could be due to the magnitudes of the local $B_z$. At the flank of the DF, different y locations determine whether the particle is in the DF or in the ambient.

Figure 7: (1) In oder to help readers understand the mechanism, it is better to also include the time history of the local $B_z$, $E_x$, $E_y$ in this figure. (2) Please provide the information in text on how to calculate the w1 and the w2. (3) The comparison between w and w1+w2 is made in Panel (d). Why do the two curves show difference after t=165s?

364-365: "Our two-dimensional Hall MHD ...... by the Hall field". This sentence is not written properly. In this study, the acceleration process is analyzed by using the test particle simulation instead of the Hall MHD model.

---

## Referee Comment (RC2) · Anonymous Referee #2 · 22 Nov 2018

The paper is devoted to the study of ion acceleration in the dipolarization front of the Earth's magnetotail. The paper focuses on the effect that an instability of the interchange kind can have in the ion energization. The authors use a test particle approach to study the acceleration process. The fields are provided by a 2D Hall MHD simulation. The authors argue that the interchange instability is responsible for ion acceleration and that the Hall electric field plays a crucial role in the process of energization and transport. In my opinion the paper contains potentially interesting results, but, before being reconsidered for publication, the authors should implement major revisions on their manuscript.

1) In order to set the value of the electric field in the simulation similar to that observed by in-situ measurements, the authors make a strong assumption on the initial condition.

[Figure]

This assumption has to be justified by physical arguments. Since the set-up is not an equilibrium, the author should provide theoretical evidences that the configuration they are considering can dynamically form, or is at least likely to be present, in the magnetotail. Moreover, I suggest that the authors plot the initial profiles of the most important quantities as a function of "x" in the case of the quasi-stationary equilibrium and in the case used for the Hall MHD simulation.

2) What boundary conditions are used for the particles? What happens to a particle that reaches the "x" or "y" boundary? Why is there an accumulation of energetic particles at y=0? This doesn't seem to be a physical effect.

3) The parameters used for the Hall MHD and the test-particle simulations must be specified. How many grid cells where used in the Hall MHD simulations? Are the electric and magnetic field coming from the Hall MHD simulation interpolated in space and time to advance particle evolution? How is this interpolation done? Which method is used for integrating the trajectories? How does the time step used to compute particle trajectories compare with the ion gyroperiod and with the time unit of the simulation? What is the direction of the test-particles initial velocity? How does the initial Larmor radius compare with the grid size?

4) In order to show an actual energization of the ions, the author should provide the PDF of particle energy at the beginning and at the end of the simulation.

5) Are the particle free to move along z? Due to the 2D field, particles do not see any field variation along z. This rules out processes such as pitch-angle scattering along Bz which can influence particle transport. The author should discuss this limitation.

The following are minor revisions.

14: The authors state that "It has been shown . . . in the magnetotail". Can they please provide a reference for this statement?

41: "SC" has not been defined previously.

54-57: "Spacecraft observations showed that . . . in the magnetotail". Either this sentence is incomplete or the word "that" has to be removed.

64: Maybe substitute "along" with "by".

70: Isn't it better to put a full stop rather than a comma after ". . . that ahead of it" ?

96-100: "Since the DF is produced by temporal . . . in the magnetotail". I don't see the connection between the sentences before and that after the comma. For example, wouldn't it be more meaningful to study this problem using a truly self-consistent PIC code?

108-109: What does it mean that ions trajectories are tracked "backward" in time?

115-117: Please explain in more details where the gravity term comes from.

125: "gx" is not contained in Equation 1.

133: Where does "p/6" come from? What is the definition of "beta"?

218-220: At what time is Figure 3 plotted?

221-231: This part on the variation of the pdf along x is kind of obscure to me. What is it meant to show?

331 (Figure 7): How are w, w1 and w2 defined?

---

## Author Comment (AC1) · 16 Jan 2019

We greatly thank the reviewer for the comments and suggestions. We have revised and improved the manuscript in response to the reviewer's comments. Following are the description of the revision we have made.

Major comments: I have trouble to understand the initial conditions of the Hall MHD simulation (Lines150-156). The authors claim that. . .

Reply: Thank you for reminding. We noticed that the dimensional units for the magnetic field and plasma velocity we chose before, $B0=15nT$, $V0=750km/s$, are too high compared with the observation values. Thus in the modified manuscript, we choose $B0=10nT$, $V0=500km/m$, which is more realistic. Under this normalization condition,

the Ey at the flank has peak value of normalized unit of 4.5 which is 22.5 mV/m. At the dawnside of the DF, Ey is 2.0 in normalized unit which is 10mV/m. The magnitude of electric field is consistent with observation results. Furthermore, the ions are mainly accclerated on the dawnside of the DF and reach energy >6keV.

As far as I understand, the test particle simulation in this manuscript is based on the electromagnetic field from a two-dimensional (2D) Hall MHD simulation. Therefore, . . .

Reply: In our manuscript, the particles, with 90-degree pitch angle and initial locations behind of the DF, move towards the earthward and dawnward and then drift to the tailDiscussion of the limitation of our model has been added in the modified manuscript.

As we know, the Hall electric field is usually formed by the relative behaviors between ions and electrons at certain spatial scales. It is very sensitive to the electon and ion kinetics. Thus, theoretically, the ion kinetic behavior due to Hall electric field is better to study by using self-consistent kinetic models. . .

Reply: The authors think the numerical is self-consistent about the ions motion, Hall electric field and the electric current as well. The ions are carrier of electric current and move along the current lines. Due to the fact that the electric current is tangential with the Dipolarization fronts, most of ions drift to the dawnside of the DF and then drift tailward at the flank. On the other hand, the Hall electric field is contributed by the tangential electric current, and has normal direction to the dipolarization fronts.

Minor comments: 19-20: "all particles". This term is not well defined. The launch information (e.g., initial energy, locations, ...) of these test particles is necessary.

Replay: In the modified abstract, a clear definition of particles we used is described. Please see in line 19-20.

26: "Ey". Please use subscript "y". Same for all the "Bx", "By", "Ex" in the text of this manuscript.

Replay: Thank you. The typo error has been corrected.

40: "magnetic flux". Does it mean "magnetic flux tube" here?

Reply: Thanks for your reminding.

41: "SC". This abbreviation has not been defined yet.

Reply: Thank you. The sentence has been corrected.

44-48: There are many possible mechanisms related to the DF generation. As a reader, I would like to know, what is the difference of the electromagnetic system of the DF generated by interchange instability than other mechanisms?

Reply: Interchange instability is considered as a possible generation mechanism for the multiple Dipolarization fronts, which have been observed in the near-Earth region in many literatures. One can imagine a picture that as a fast Earthward flow approaches the Earth, it would be braked by the ambient plasma. In the braking region, the tailward gradient of thermal pressure increases and meanwhile the earthward magnetic curvature force increase, which consequently leads to a tailward force. This tailward force enables Earthward fast flows decelerate initially and brake finally in the near Earth plasma sheet. In conjunction with the tailward gradient of plasma density due to the flow braking, the total force brings forth interchange instability in the braking region.

108: "ions trajectories" ! "ion trajectories"

Reply: Thank you. The typo error has been corrected in line 109.

108: "track ions trajectories backward in time". This phrase makes a little confusion. It could mean either running the simulation with negative time step from later time to earlier time or running the simulation with positive time step then check the time history of the trajectories of selected particles. Please specify which scenario is used in this manuscript.

Reply: Thank you. Actually we adopted the latter in our manuscript.

111: "realistic". This word is not used properly. The simulation in this study is only two

dimensional and the parameters are not set up based on spacecraft observations for specific events.

Reply: In our manuscript (l.145 - 150), we pointed out that the simulation electric field is too small in the previous MHD simulation. The "realistic" refers to the more realistic electric field.

122: (1) "$\mu\_0$" doesn't look correct in the second and fourth equations. (2) The "g" should be vector "g". ...

Reply: Thank you. The typo error has been corrected.

124: "$\beta$". There is no an explicit "$\beta$" shown before this sentence.

Reply: Thank you. "$\beta$" should be included in the expression of "g", which has been added in the modified manuscript.

125: "g_x". There is no an explicit "g_x" shown before this sentence. ...

Reply: Thank you. The missing statements have been added.

129: What is the value of "$\beta$" used?

Replay: $\beta$=2 is plasma beta.

138: According to Guzdar et al. (2010), there is a missing "" on the left hand side of the equation.

Reply: The typo error has been corrected in the modified manuscript.

145: "we set" should be "they set".

Reply: It has been corrected in the modified manuscript.

154-155: What are the values for _L and _R?

Reply: In our manuscript, we took _L=1.5 and _R=1.

158-159: (1) "The simulation box is 2 RE and 1.5 RE in the direction of x and y". This description doesn't look correct. The simulation results (i.e., the figures) show that there is 4RE in x. (2) Since the simulation used in this study is two-dimensional, please mention this information in the Abstract and somewhere in the "Theoretical and Numerical Model" section. (3) Please provide the information about the number of grid cells in each dimension.

Reply: Thank you. In our simulation, the simulation box is 4 RE and 1.5 RE in the direction of x and y. The numbers of grid cells in x and y directions are set 301 and 201, respectively. The error has been corrected.

161-167: As a reader, I would like to know more details about the test particle simulation: (1) Time step (2) The minimum gyration period of the simulated ions (3) Time integration scheme (4) What happens if the test particle hits the boundary of the simulation box? (5) Since the electromagnetic field is from a two-dimensional Hall MHD simulation, is the test particle simulation also two-dimensional (i.e., do not consider the motion along the z direction)? If so, the test particles should only represent the particles with 90-degree pitch angle.

Reply: In our simulation, we adopt the MHD time step as the test particle simulation time step. At each time step, we use the fourth order Runge-Kuta to solve the equation of motion. Once the test particle hits the boundary, we assume that the ion will remain stationary at the boundary.

172: "more realistic". It is not clear in what aspects this is more realistic. Please specify.

Replay: Compared with previous MHD simulation, we improve the initial condition to obtain a higher electric field. So, the "more realistic" respects the value electric field.

176: "DF". The DF is commonly defined by the abrupt increase of Bz component. However, Figure 1 doesn't show any information about Bz. Please point out in Figure 1 where is DF and how it is defined.

Replay: Actually, the boundary of Bz component abrupt increase is the electric field boundary in Figure1 and we define the DF as the front boundary of Bz component abrupt increase.

182-184: "It should be ...... with a stronger dawnside electric field." It is not clear which asymmetry is mentioned in this sentence. Which electric field component? What time and location in Figure 1?

Reply: From Figure 1 (a), (d) we can clearly see that the dawnward electric Ex, Ey are both larger than the duskward at the region -1 RE < x < 1RE.

184-187: Please specify the locations with either Hall electric field or convective electric field . . .

Replay: Thank you! The asymmetry of distribution of electric field is a subsequence of Lorentz force, according to the Hall electric field, EHall=J×B/ne. Consequently, the Lorentz force along the tangent plane of DF associated with the motions of the decoupled ions leads to the asymmetry of the "mushroom" pattern (see Lu et al. 2013).

192: "simulation box". Should it be "launch region of the test particles", since the results show that some of the particles are located outside this box? In addition, please also provide the DF location in text related to this launch region.

Reply: Thank you. The simulation box is x = -2 RE ∼ 2 RE, y = -1.5 RE ∼ 0 RE and the launch region is x = -0.9 RE ∼ -0.4 RE, y = -1.46 RE ∼ -0.04 RE. In Figure2 black line represents the position of the DF.

196: "protons". In this manuscript, both "proton" and "ion" are used. Since there is no any other ion species than proton, it should be better to only use one of them.

Reply: Thank you. In the modified manuscript, only "ion" is used.

197: "black lines indicate the position of DFs". Are the black lines schematic or calculated? What variable is used to determine the black lines? Same comments for Figures

4 and 5b. In addition, what does the Bz profile look like in the launch region? It would be helpful for comparing with observations if there is a line plot of Bz as a function of x.

Reply: Actually the black lines are schematic and we define the black lines by the boundary of the abrupt increase of the Bz component. The Bz profile in the launch region can be shown in Figure I.

Figure 2: (1) (Same for Figure 4) What happens to the particles hitting the simulation boundary in y direction? Are they still moving (e.g., some energetic particles at y=0 in (b) is not there any more in (c)) (2) Why are there more energetic particles in earlier time (b) than those in later time (c)? Do they lose energy?...

Reply: The ions will remain stationary at the boundary once they hitting the simulation boundary in y direction. In our manuscript the particle energy refers to the instantaneous energy, so the energy changes with time.

Figure 3: (1) What time is this plot taken? (2) It should be better to add the initial PDFs so that the initial power law can also be compared. (3) What are the areas to select the particles for plotting each line at different x location? Do these PDF curves include the particles at the y boundary? How much percentage of the energetic particles are at the boundary? If the number is significant, it should be better to perform a simulation with a bigger box.

Reply: In our manuscript, Figure 3 plot the PDFs of total particle energy at t = 187s. The initial power law energy distribution $F\sim(1+h/(\kappa T\_0 ))^{(-\kappa-1)}$, which is similar to kappa distribution.

223: "among the multiple fold lines". It is not quite clear what it means. Please rephrase it.

Replay: Thank you. We have rephrased the expression in line 229-231.

226: "small"! $\rightarrow$ "low"

Reply: Thank you. It has been modified.

247-248: "they move earthward and dawnward with a larger gyration radius due to smaller ambient magnetic Bz." This is about the ions with initial positions ahead of the front. The dawnward motion is not explained well. Is the gyration radius comparable to the scale of the DF? Is the grad-B drift due to the set-up of the Hall MHD magnetic field considered?

Reply: Actually the ions move dawnward because of the Hall electric field in the –x direction. The ion gyration radius is larger than the scale of the DF.

277: "distribution of differential energy flux". How large area is used to select the particles for the calculation? In addition, please change the label of Figures 5c and d to "differential energy flux" instead of "flux".

Reply: We chose 0.1RE*0.1RE as the statistical calculation area and the expression has been polished in the modified manuscript.

278: "azimuthal angle". Which direction is indicated by the zero degree? Please define the directions indicated by different azimuthal angles.

Reply: As shown in the following picture.

279-280: "Ion trajectories with initial positions along different y distances". The meaning of this sentence is not quite clear (e.g., the y distance from where?), please rephrase it. In addition, it is very hard to obtain information from Figure 5b because too many lines overlapping to each other (e.g., can't find the initial position, can't follow individual trajectory, ...).

Reply: Thank you. The ambiguity has been corrected.

286-290: "At about t=146 ...... they are almost simultaneously observed (Figure 5b and 5d)". The meaning of this sentence is not clear, please rephrase it. In addition, why is there a gap around t=153s in Figures 5c and d?

Reply: The ions ahead of the DF accelerated first and the behind one can't be accelerated until they reach the front of the DF. So there exit a gap between these two particles.

Figure 6: Figure 7 shows that the particle obtains large amount of energy from w2 between t=163-165s. It should be better to also show Ey during this time period in Figure 6.

Reply: Thank you. We have added Figure 8 to the manuscript.

316-320: The explanation of the gradual increase of the kinetic energy is not clear. Because the magnitude of $\delta\hat{y}$- and $\delta\hat{y}$+ could be due to the magnitudes of the local Bz. At the flank of the DF, different y locations determine whether the particle is in the DF or in the ambient.

Reply: when t > 166 s, the ion kinetic energy gradually increases, which can be interpreted based on the fact that the y-displacement $\delta\hat{y}$+ (corresponding to the energy increase) is larger than $\delta\hat{y}$- (corresponding to the energy reduction) in the case where Ey component is almost constant. Since the ions arrive at the ambient of dawnward DF, the magnitude of magnetic field is increase, which results in a high $\delta\hat{y}$+.

Figure 7: (1) In order to help readers understand the mechanism, it is better to also include the time history of the local Bz, Ex, Ey in this figure. (2) Please provide the information in text on how to calculate the w_1 and the w_2. (3) The comparison between w and w1+w2 is made in Panel (d). Why do the two curves show difference after t=165s?

Reply: Thank you. Figure 8 has been added in the modified manuscript to show the time history of the local Bz, Ex, Ey in Figure 7. Since we used the formula w_1=$\Delta$E_x·$\Delta$x and w_2=$\Delta$E_y·$\Delta$y to calculate w_1 and w_2, so there should be computational error compared to the definition of kinetic energy of particles.

364-365: "Our two-dimensional Hall MHD ...... by the Hall field". This sentence is not

written properly. In this study, the acceleration process is analyzed by using the test particle simulation instead of the Hall MHD model.

Reply: Thank you. We have corrected the wrong sentence.

Please also note the supplement to this comment:
https://www.nonlin-processes-geophys-discuss.net/npg-2018-43/npg-2018-43-AC1-supplement.pdf
* * *
[Figure]

**Fig. 1.**

[Figure]

**Fig. 2.**

---

## Author Comment (AC2) · 16 Jan 2019

We greatly thank the reviewer's helpful comments and suggestions on our manuscript, which are very useful for us to improve our manuscript. Following are the reply to the suggestions.

Major comments: In order to set the value of the electric field in the simulation similar to that observed by in-situ measurements, the authors make a strong assumption on the initial condition. This assumption has to be justified by physical arguments. Since the set-up is not an equilibrium, the author should provide theoretical evidences that the configuration they are considering can dynamically form, or is at least likely to be present, in the magnetotail. Moreover, I suggest that the authors plot the initial profiles

of the most important quantities as a function of "x" in the case of the quasi-stationary equilibrium and in the case used for the Hall MHD simulation.

Reply: The initial condition used in our manuscript is based on one equilibrium equation. We plot the $\_0, B\_z$ and $E$ along the $y = -0.6$.

2) What boundary conditions are used for the particles? What happens to a particle that reaches the "x" or "y" boundary? Why is there an accumulation of energetic particles at $y=0$? This doesn't seem to be a physical effect.

Reply: As for the boundary condition for the ions, they remain stationary at the boundary once they hitting the simulation boundary in y direction. In our manuscript there are two vortex flow pattern as a consequence of the earthward flows coexisted with the tailward flows. The particles are concentrated at $y=0$ because of the vortex flow pattern.

3) The parameters used for the Hall MHD and the test-particle simulations must be specified. How many grid cells where used in the Hall MHD simulations? Are the electric and magnetic field coming from the Hall MHD simulation interpolated in space and time to advance particle evolution? How is this interpolation done? Which method is used for integrating the trajectories? How does the time step used to compute particle trajectories compare with the ion gyroperiod and with the time unit of the simulation? What is the direction of the test-particles initial velocity? How does the initial Larmor radius compare with the grid size?

Reply: Thank you. The numbers of grid cells in x and y directions are set 301 and 201, respectively. The magnetic and electric fields of ions were calculated by PIC (Particles in Cell) method. In our simulation, we adopt the MHD time step as the test particle simulation time step. At each time step, we use the fourth order Runge-Kuta to solve the equation of motion. The direction of the initial velocity of the ion is random. Under the condition we chose, the initial Larmor radius is 522km which is 0.082 in dimensionless unit. It occupies 6 and 11 grid points in x and y direction respectively.

4) In order to show an actual energization of the ions, the author should provide the PDF of particle energy at the beginning and at the end of the simulation.

Reply: The initial power law energy distribution F$\sim$(1+h/($\kappa$T_0 ))^(-$\kappa$-1), which is similar to kappa distribution.

5) Are the particle free to move along z? Due to the 2D field, particles do not see any field variation along z. This rules out processes such as pitch-angle scattering along Bz which can influence particle transport. The author should discuss this limitation.

Reply: Thank you. The corresponding context has been added in the modified manuscript.

14: The authors state that "It has been shown . . . in the magnetotail". Can they please provide a reference for this statement?

Reply: It has been modified in the new manuscript. 41: "SC" has not been defined previously.

Reply: It has been modified in the new manuscript.

64: Maybe substitute "along" with "by".

Reply: Thank you.

70: Isn't it better to put a full stop rather than a comma after ". . . that ahead of it"?

Reply: Thank you.

96-100: "Since the DF is produced by temporal. . . in the magnetotail". I don't see the connection between the sentences before and that after the comma. For example, wouldn't it be more meaningful to study this problem using a truly self-consistent PIC code?

Reply: The sentence has been rephrased in the modified manuscript.

108-109: What does it mean that ions trajectories are tracked "backward" in time?

Reply: In the simulations, we run the simulation with positive time step then check the time history of the trajectories of selected particles.

115-117: Please explain in more details where the gravity term comes from.

Reply: Interchange instability is considered as a possible generation mechanism for the multiple Dipolarization fronts, which have been observed in the near-Earth region in many literatures. One can imagine a picture that as a fast Earthward flow approaches the Earth, it would be braked by the ambient plasma. In the braking region, the tail-ward gradient of thermal pressure increases and meanwhile the earthward magnetic curvature force increase, which consequently leads to a tailward force. This tailward force enables Earthward fast flows decelerate initially and brake finally in the near Earth plasma sheet. In conjunction with the tailward gradient of plasma density due to the flow braking, the total force brings forth interchange instability in the braking region.

125: "gx" is not contained in Equation 1.

Reply: The typo error has been corrected in the modified manuscript.

133: Where does "p/6" come from? What is the definition of "beta"?

Reply: Electron pressure pe is taken as p/6, because the proton temperature is 5 times that of electron temperature. $\beta$ is plasma beta.

218-220: At what time is Figure 3 plotted?

Reply: Figure 3 plot the PDFs of total particle energy at t = 286s.

221-231: This part on the variation of the pdf along x is kind of obscure to me. What is it meant to show?

Reply: We plot Figure3 to get a better sense of the distribution of the ions.

(Figure 7): How are w, w1 and w2 defined?

Reply: We used the formula $w_1=\Delta E_x \cdot \Delta x$ and $w_2=\Delta E_y \cdot \Delta y$ to calculate $w_1$ and w_2. W is the kinetic energy of the particle given by 1/2 mv^2.

Please also note the supplement to this comment:
https://www.nonlin-processes-geophys-discuss.net/npg-2018-43/npg-2018-43-AC2-supplement.pdf
* * *
[Figure]

**Fig. 1.**

**Supplement:**

[revised manuscript text omitted]

5 mV/m.

The dimensionless model with an effective gravity is as follows:

$$
\frac{\partial}{\partial t}\begin{bmatrix} \rho \\ \rho\mathbf{U} \\ \mathbf{B} \\ \rho e_t \end{bmatrix} + \nabla\cdot \begin{bmatrix} \rho\mathbf{U} \\ \rho\mathbf{U}\mathbf{U} + P\mathbf{I} - \dfrac{\mathbf{B}\mathbf{B}}{\mu_0} \\ \mathbf{U}\mathbf{B} - \mathbf{B}\mathbf{U} \\ (\rho e_t + P)\mathbf{U} - \dfrac{\mathbf{B}}{\mu_0}(\mathbf{U}\cdot\mathbf{B}) \end{bmatrix} = \begin{bmatrix} 0 \\ \mathbf{g} \\ 0 \\ \mathbf{g}\cdot\mathbf{U} \end{bmatrix} +
$$

$$
d_i \begin{bmatrix} 0 \\ 0 \\ -\dfrac{1}{\mu_0}\nabla\times\left(\dfrac{\nabla\times\mathbf{B}\times\mathbf{B}}{\rho}\right) \\ -\dfrac{1}{\mu_0^2}\mathbf{B}\cdot\left[\nabla\times\left(\dfrac{\nabla\times\mathbf{B}\times\mathbf{B}}{\rho}\right)\right] \end{bmatrix} + d_i \begin{bmatrix} 0 \\ 0 \\ -\dfrac{1}{\mu_0}\nabla\times\left(\dfrac{\nabla p_e}{\rho}\right) \\ -\dfrac{1}{\mu_0^2}\mathbf{B}\cdot\left[\nabla\times\left(\dfrac{\nabla p_e}{\rho}\right)\right] \end{bmatrix}
$$

(1)

Where $P = p + \mathbf{B}^2/2\mu_0$, $\mathbf{U}$ and $\mathbf{B}$ are velocity vector and magnetic field vector, respectively, $\rho e_t = \rho\mathbf{U}^2/2 + p/(\gamma - 1) + \mathbf{B}^2/2\mu_0$ ,

$\mathbf{g} = [\beta\rho g_x/2, 0, 0]^T$ , $\beta$ is plasma beta, $g_x$ is the effective gravitational force in x direction. In equation (1), the second and third terms on the right-hand side represent the Hall effect and the electron pressure gradient, respectively. In our present numerical cases, we postulate that plasma is under isothermal conditions with an isothermal equation of state $p = \beta\rho/2$ and take the adiabatic exponent $\gamma = 5/3$.

The ion inertial length $d_i = (m_i/\mu_0 e^2 Z^2 L^2 n_i)^{1/2}$ , given the reference length L = 1 $R_E$, the dimensionless ion inertial length is taken as

$d_i \approx 0.1$. Electron pressure $p_e$ is taken as $p/6$, because the ion temperature is 5 times that of electron temperature (Baumjohann et al.,

1989; Artemyev et al., 2011).

As for initial conditions, the quasi-stationary equilibrium built by the plasma pressure and effective gravity g (see equation (2)) (Guzdar et al.

2010 and Lu et al. 2013, 2015) is theoretically reasonable.

$$\hat{g}\frac{\beta}{2}\rho = \frac{\partial}{\partial x}\left(\frac{\beta}{2}\rho + \frac{B_z^2}{2}\right) \tag{2}$$

It should be noticed that the dawn-dusk and earthward electric field components averagely, increase to ~5 mV/m along with the transient Bz increase and in some events, the electric field increase exceeded 10

mV/m (Runov et al., 2009, 2011; Schmid, D., et al. 2016). However, the electric fields calculated by the Hall MHD model in Lu et al. (2013) are smaller than the observations (see Lu et al., 2013, for a typical dipolarization event at x = -10 $R_E$ in the equatorial plane, they set $B_0$ =

15 nT, leading to Bz changed from 10.2 nT to 16.8 nT after DF

propagation. The electric field components $E_x$ and $E_y$ are both less than

3 mV/m). So, it is reasonable that we improve the initial conditions to obtain a realistic electric field, which plays a vital important role in ion energization.

We take the initial conditions as follows:

$$\rho(\text{x}) = \frac{1}{2}(\rho_L + \rho_R) - \frac{1}{2}(\rho_L - \rho_R)\tanh\left(\frac{x}{l}\right) \tag{3}$$

$$\quad \begin{cases} B_Z(x) = 0.28x + 0.7535 & (x \le -0.38) \\ B_Z(x) = 1.5 + \tanh\left(\frac{x}{0.3}\right) & (-0.38 < x < 0.4) \\ B_Z(x) = 0.14x + 2.314 & (x \ge 0.4) \end{cases} \quad (4)$$

Given the generalized Ohm's law, we use a piecewise function to describe Bz so as to obtain a strong electric field. In equation (3),

$\rho_L = 2$ and $\rho_R = 1$ are the density closer to and away from the Earth, respectively and the characteristic scale l = 0.2 $R_E$. The numbers of grid cells in x and y directions are set 301 and 201, respectively.

We solved equation (1) by adopting the second-order upwind total variation diminishing scheme. The simulation box is 4 $R_E$ and 1.5 $R_E$ in the direction of x and y, respectively. The x boundary is assumed to be zero for all perturbed quantities and the y boundary is to be periodic.

As the second simulation step, the control equations for ion motion should be given. Typically, the drift approximation breaks down in terms of ion motion in magnetotail. The dimensionless equations of motion are given by

$$\quad \begin{cases} \frac{d\boldsymbol{r}}{dt} = \boldsymbol{u} \\ \frac{d\boldsymbol{u}}{dt} = \alpha(\boldsymbol{E} + \boldsymbol{V} \times \boldsymbol{B}) \end{cases} \quad (5)$$

where $\boldsymbol{r}$ is the particle position, $\boldsymbol{u}$ is the particle velocity, the dimensional parameter $\alpha = \frac{1}{d_i} \approx 10$.

**Simulation Results**

From 0s to 221s, the simulation experienced a pre-onset phase, during which the DF formed as a consequence of effective gravity g interaction with plasma density gradient. In order to be more realistic, we set up the time interval from 221s to 286s as the acceleration period of the particles.

Figure 1 shows the evolution of the electric field in the z = 0 plane and black lines indicate streamlines, one can clearly see that the DF moves toward the Earth as time passes by. From Figure 1b it can be seen that the earthward flows coexisted with the tailward flows of the dawn and dusk edges, as a consequence two vortex flow pattern appeared. Figure 1 also shows that the electric field components $E_x$ and $E_y$ are both normal to the front, which is consistent with the observation and simulation (Fu et al.,

2012b; Lu et al., 2013). It should be noticed that the total electric field E

is asymmetrically distributed on the DF, with a stronger dawnside electric field in Figure 1. This asymmetry can be interpreted as a subsequence of

Lorentz force, according to the Hall electric field. Consequently, the

Lorentz force along the tangent plane of DF associated with the motions of the decoupled ions leads to the asymmetry of the "mushroom" pattern (Lu et al. 2013).

[Figure]

**Figure 1.** Evolution of the electric field $E_x$ (a-c) and $E_y$ (d-f), black line in (b) indicate streamlines

At t = 144.5s, we numerically distribute test particles (80000 ions in total)

around the DF (the launch region with x = -0.9 $R_E$ ~ -0.4 $R_E$, y = -1.46 $R_E$

~ -0.04 $R_E$) with the initial power law energy distribution $F \sim (1 +$

$h/\kappa T_0)^{-\kappa-1}$(we take $\kappa = 5,\ T_0 = 1.5\ keV$ and h from 1 keV to 10 keV)

( Artemyev et al., 2015). Figure 2 exhibits the spatial distribution of ions at a given moment. The energy of particles is marked with color and black lines indicate the position of DFs. As time passes by, the ions behind the DF accelerate and transport to the dawn flank of the DF, resulting in the reduction of the ion density behind the DF. It should be pointed that the ions will remain stationary at the boundary once they move to the simulated boundary. 
[revised manuscript text omitted]
 and Figure 8 shows the local $B_z$, $E_x$, $E_y$ seen by this particle. During the beginning period from 221s to 247s, the ion moves earthward together with the front and meanwhile dawnward in the frame of the moving front. During this period, the ion gains very little energy. Even though the $E_x$ component of electric field accelerates the ion along its earthward motion, the deceleration by the $E_y$ component keeps the ion energy almost unchanged. When t = 249.6s, the ion arrives at the dawnside of the DF, where the Hall electric field is very strong. After a sharp energization for about 5 seconds, the ion kinetic energy increase to ~ 5 keV (Figure 7b and 7c, the weaker $E_x$ works to reduce the energy by about 3.7 keV and the stronger $E_y$ increases the energy by about 9 keV). As shown in Figure 6 and 7, when t > 253.8 s, the ion kinetic energy gradually increases, which can be interpreted based on the fact that the y-displacement $\delta y^+$ (corresponding to the energy increase) is larger than $\delta y^-$ (corresponding to the energy reduction) in the case where $E_y$ component is almost constant. Since the ions arrive at the ambient of dawnward DF, the magnitude of magnetic field is increase, which results

    in a high $\delta y^+$.Figure 8 shows the time history of the local Bz, $E_x$, $E_y$.

[Figure]

**Figure 6.** Orbits of an ion with the initial energy 1 keV and final energy

5.54 keV, traced from x = -0.7 $R_E$, y = -0.86 $R_E$ at different moments. The locations of ion are shown as red dots superposed on snapshots of the background Hall electric field $E_x$ (a-b) and $E_y$ (c-f).

[Figure]

**Figure 7.** Physical quantities of ion as the function of time with blue dotted lines index specific moment. (a), (a') Y position and its partial enlarged detail, red dotted line is the reference line. (b, c) Energization produced by $E_x$ and $E_y$, respectively. (d) Kinetic energy and numerical summation of w1 and w2 display with orange and black line, respectively. The label of t1 to t5 correspond to 249.6s, 252.2s, 254.8s, 256.9s and 260s respectively.

[Figure]

**Figure 8.** The time history of the local Bz, $E_x$, $E_y$

**Summary and Discussion**

In this paper, we used a test particle simulation to investigate ion acceleration at dipolarization fronts (DFs) produced by interchange instability in the magnetotail. The Hall MHD model was improved by applying the realistic initial conditions to obtain the fields which are better consistent with observation.

It should be noticed that our test particle is 2D without the motion in the z direction along the field line. So we only study the ions with 90-degree pitch angle. 
[revised manuscript text omitted]